# Peste des Petits Ruminants (PPR) Vaccination Cost Estimates in Burkina Faso

**DOI:** 10.3390/ani12162152

**Published:** 2022-08-22

**Authors:** Guy Sidwatta Ilboudo, Papa Abdoulaye Kane, Pacem Kotchofa, Edward Okoth, Adama Maiga, Michel Dione

**Affiliations:** 1International Livestock Research Institute, Ouagadougou 01 BP 1496, Burkina Faso; 2Institut Sénégalais de Recherches Agricoles, Bureau d’Analyse Macroéconomique, Dakar BP 3120, Senegal; 3International Livestock Research Institute, Ouakam, Dakar BP 24265, Senegal; 4International Livestock Research Institute, Nairobi P.O. Box 30709-00100, Kenya; 5Direction Générale des Services Vétérinaires, Ouagadougou 09 BP 907, Burkina Faso

**Keywords:** peste des petits ruminants, small ruminants, livelihood, vaccination strategy

## Abstract

**Simple Summary:**

Peste des petits ruminants (PPR) causes high mortality in sheep and goats leading to negative social, cultural, and economic impacts on farmers who keep small ruminants. Since 2019, Burkina Faso has been implementing a national strategy to eliminate PPR. After two years of mass vaccination of small ruminants with significant resources invested, very little is known about the cost of vaccination and how it is distributed along with the different nodes of the vaccine distribution chain. This study aimed to fill this gap to inform decision-making in the allocation of the limited resources that are available. The results show that the cost of vaccination of a small ruminant is XOF 169 (USD 0.3) and XOF 103 (USD 0.18) if the vaccination is carried out by public and private vaccinators, respectively. Field activities and personnel bear the highest cost share. These results will inform a better resource allocation to improve the effectiveness and efficiency of small ruminants vaccination against PPR.

**Abstract:**

Every year the government organizes country-wide vaccination campaigns targeting peste des petits ruminants (PPR) for small ruminants (sheep and goats). Despite the heavy investment in vaccination, no study has either rigorously estimated or described the cost of vaccine delivery. This study aimed to fill this gap by assessing and describing the cost of delivery of vaccines against PPR using the 2020 vaccination campaign data. The microcosting approach based on the World Health Organization (WHO) guidelines to construct comprehensive multiyear plans (cMYP) for human immunization programs was used. The cost data is presented for the public and private vaccine delivery channels separately and analyzed using three approaches considering activity lines, inputs, and nature of cost (fixed versus variable). Results show that the unit cost of vaccinating a sheep or goat is estimated at XOF 169 (USD 0.3) and XOF 103 (USD 0.18) through the public and private channels, respectively. Using the activity line framework, we found that the field activities including charges for vaccinator, cost of vaccination materials, and field transportation account for more than 50% of the vaccination cost. In terms of inputs, the personnel cost is the most significant contributor with 65%. Fixed costs are higher in the public sector with up to 46% compared to the private sector which is estimated to take 26% of the cost. This study informs veterinary services’ investment decision options for a better allocation of resources in implementing PPR and other small ruminant disease control efforts in Burkina Faso and the Sahel.

## 1. Background

Small ruminants (SR) are a very important component of livestock production globally, particularly in developing countries. The global annual SR production is estimated to be around 11 million tons of meat and 22 million tons of milk with more than 78% of the total production coming from Africa and Asia [1,2]. With more than 15 million goats and 10 million sheep in 2018, SR play an important economic, social, and cultural role in pastoral communities and contributes strongly to food security and nutrition in Burkina Faso [3,4,5,6]. However, SR production faces multiple constraints, particularly related to health. Peste des petits ruminants (PPR) is one of the most fatal diseases of SR in Burkina Faso [6]. PPR is a highly contagious, acute infectious disease caused by a Morbillivirus of the Paramyxoviridae family and related to the viruses responsible for rinderpest, measles, and distemper [7]. It is characterized by fever, discharge from the eyes and nose, erosions in the mouth, bronchopneumonia, and diarrhea [8]. The severity of clinical signs, morbidity rate, and case fatality rate may vary depending on the virulence of the virus strain, the species and breed type of the host, the concomitant infection, and the exposure of the animal population prior to infection [7]. When introduced into a naive population, morbidity and mortality can reach almost 100%, causing a major shock to the livelihoods of pastoralists and the SR trade. When uncontrolled, the disease becomes endemic, and infection persists in the SR populations resulting in productivity losses that leads to long-term negative impact on the poorest and most marginalized ranching households, with a huge impact on women [9] who mostly keep the animals. The disease can be easily controlled with vaccination using a live attenuated virus vaccine of the Nigerian 75/1 strain. The vaccine provides protective immunity to the animal for at least 3 years [10]. However, efficient delivery of the vaccine has always been a big challenge in Burkina Faso as it is the case in all Sahelian countries because of many reasons including poor maintenance of the cold chain needed to maintain the integrity of the vaccine, a low level of awareness by farmers of benefits of vaccination, and the poor capacity of veterinary services to support logistics associated with vaccine delivery, amongst others [11,12]. The economic impact of PPR was estimated between 2014 and 2017 at more than XOF 285 billion (USD 490 million) due to mortalities and morbidities in Burkina Faso [13]. A modeling study in the country reported that considering a 5% shock in the value of SR output to simulate the hypothetical outbreak would reduce the Gross Domestic Product (GDP) at factor cost by 0.62% (i.e., over USD 98 million) considering the GDP in 2019. It would further cause a contraction in the value of SR by 5% while reducing maize and rice production value by over 0.5% and 0.65% for the other cereals. About 0.39% of all jobs (i.e., about 22,000 jobs) would be lost with losses concentrated in the SR sector and across various crop production sectors, including maize (0.49%), rice (0.51%), and other cereals (0.59%) [14]. Because of its economic and health importance, PPR has been ranked among the priority diseases to be targeted for strict control in Burkina Faso [6,15].

In 2015, the international community set the goal of eradicating PPR by 2030, and, since then, the Food and Agriculture Organization (FAO) (Rome, Italy) and the World Organization for Animal Health (WOAH) (Paris, France) have developed and are implementing a Global Control and Eradication Strategy (GCES) for PPR. Vaccination is the main component of this strategy. Like other countries, Burkina Faso, implemented its second year of PPR control and eradication plan in 2020 [16,17]. This plan mainly includes an annual mass vaccination of sheep and goats aged at least three months. The vaccination campaign is implemented through a public–private partnership (PPP) involving many stakeholders such as government, private veterinarians, farmers, and local leaders for around three months.

Theoretically, annual mass vaccination is an effective control measure, but in practice, it is difficult to achieve and is expensive [9]. Indeed, the implementation of vaccination campaigns requires the mobilization of significant human, material, and financial resources [16]. In addition, there is a need to mobilize farmers for massive adherence to vaccination in a context of poverty and livestock mobility in time and space [6,11,16].

Despite the support of various projects such as “Projet régional d’appui au pastoralisme au Sahel” (PRAPS) [18] and “Projet d’appui au développement du secteur de l’élevage au Burkina Faso” (PADEL-B) [19], it must be recognized that the vaccination coverage of SR against PPR is not sufficient to eradicate the disease. For example, in 2019 and 2020, a total of 2.4 million (9% coverage) and 4.4 million (17% coverage) sheep and goats were vaccinated, respectively [4,20]. Most of the time, the allocation of resources is not adequate and does not always meet the real needs of the stakeholders. This often poses a problem of efficiency and results in low vaccination coverage. To date, no study has rigorously estimated and described the comprehensive cost of the country’s PPR vaccine delivery. This study, therefore, aimed to determine the cost structure along the livestock vaccine distribution chains using PPR in SR as an example, and identify the cost variations depending on the sector (public versus private), the types of activities, the types of inputs to vaccination, and the nature of the costs (fixed versus variable). This will enable us to formulate recommendations for an efficient allocation of resources to guarantee the effectiveness of the PPR control campaigns.

## 2. Materials and Methods

### 2.1. Study Location

Data was collected from most parts of the country except areas facing high insecurity. Consequently, 6 out of the 13 regions of the country including Boucle du Mouhoun, Cascades, Centre-Est, Centre-Ouest, Hauts Bassins, and Plateau Central were enrolled (Figure 1). The regions were selected purposively to be geographically representative based on the distance to the central veterinary services located in Ouagadougou, the capital city. Three regions (Boucle du Mouhoun, Hauts Bassins, and Cascades) are located 250 km, 350 km, and 450 km, respectively, far from central veterinary services; while the three other regions (Plateau Central, Centre-Ouest, and Centre-Est) are nearer to the central veterinary services, located at 30 km, 100 km, and 180 km, respectively. In each selected region of the country, one representative province was purposively selected and, accordingly, Leraba, Tuy, Ziro, Kouritenga, Kourweogo, and Banwa were selected, making a total of six provinces. Subsequently, in each selected province, one commune was randomly selected and, accordingly, Kankalaba, Boni, Bognounou, Tensobentenga, Toeghin, and Kouka were selected, making a total of six communes.

### 2.2. Data Collection

The data was collected from November 2020 to January 2021 using several methods.

Desk review: published and unpublished reports on PPR control strategies [4,16,20] and vaccination evaluation campaigns from previous years (2013 and 2018) [3,12] were consulted either online or at the central veterinary services.Key informant interviews (KII): Informal interviews were held with five officials from the central veterinary services at the Ministry of Livestock (Ministère des Ressources Animales et Halieutiques-MRAH) (Ouagadougou, Burkina Faso) including the director of the veterinary service, the director of animal health, and staff from the service in charge of livestock vaccination. Two field officers from the public veterinary services, two private veterinarians, and two staff for the administration and finance office of the MRAH were also interviewed. To obtain data on the cost of electricity, two staff from the National Electricity Agency (Société Nationale d’Electricité du Burkina Faso—SONABEL) (Ouagadougou, Burkina Faso) were interviewed.Structured individual interviews: A total of 43 people from institutions involved in livestock vaccination were interviewed using a questionnaire. They included 7 central veterinary services staff (including those leading strategic project in livestock such as “Projet d’Appui au Développement du secteur de l’Elevage au *Burkina* Faso”- PADEL-B and “Projet Régional d’Appui au Pastoralisme au Sahel”- PRAPS), 6 regional directors, 12 provincial directors, 12 public field staff (vaccinators), and 6 private veterinarians.Tool validation: the data collection tools were presented to stakeholders from the central veterinary services for their input and validation prior to rolling out in the field to make sure that important vaccination cost items were not missed during field data collection.Validation workshop: The preliminary results were validated by stakeholders during a national workshop in Ouagadougou at the livestock ministry. The workshop was attended by35 participants made up of MRAH representatives, central veterinary services staff, PADEL-B and PRAPS representatives, regional directors, provincial directors, public vaccinators, private veterinarians, private veterinarians association (Collectif des Vétérinaires Privés—COVEP) (Ouagadougou, Burkina Faso), National Veterinarians Order (Ouagadougou, Burkina Faso), and International Committee of the Red Cross (ICRC) (Geneva, Switzerland).

### 2.3. Costing Model

We used the microcosting approach based on the World Health Organization (WHO) (Geneva, Switzerland) guidelines which constructed comprehensive multiyear plans (cMYP) for human immunization programs vaccine [21]. The cost calculation methods allowed the outputs to be presented in three ways: 

Cost by activity: the different types of activities carried out at each level of the vaccine supply chain during the vaccination campaign including vaccine purchase, transportation, storage, field delivery, sensitization, training, meetings, supervision, and coordination.Cost by inputs: the main inputs to vaccination including the vaccine (product), personnel, material and logistics, vaccine wastage, and overheads.Cost by nature of the cost: fixed versus variable costs.All component costs were calculated at each level, the private and the public channel being separated

#### 2.3.1. Calculation Methods

Data were entered and analyzed in Excel sheets (Microsoft corporation). The average values were calculated for each level of vaccine distribution channel. The calculation of the different costs was made based on the methods listed below and also by rules-of-thumb [21].

The past spending approach was used: The 2020 PPR vaccination campaign data was also used. A supplemental cost such as the vaccination campaign launching ceremony of 2019 was added; in 2020, this activity did not take place due to Covid19 pandemic restrictions.The ingredient approaches and shared cost method:

*TCi* = (*Qi* × *Pi*) *∅*,(1)
with *∅* representing the percentage of (*i*) used for PPR vaccination. *TC* is the total estimate of the (*i*). The quantity (*Q*) is the number of units of an item used. The price (*P*) is the price of the cost item.

The unit costs were calculated for all cost items divided by the number of animals vaccinated at the correspondent level:*Unit cost* = *Total cost/number of animal vaccinated*(2)

The capital or fixed costs (costs that vary depending on the number of the animals vaccinated) were calculated using the straight-line depreciation method:*Material depreciation* = *Material cost/ULY*(3)

The useful life years (*ULY*) were used considering the international value.

Following the WHO guidelines, the cold chain equipment maintenance cost was estimated to be 2.5% of the new cold chain equipment price. The mean of transportation maintenance cost was estimated to be 15% of the fuel cost for the same activity [21].

#### 2.3.2. Cost Calculation by Type of Activities

The details of the calculation by activities are presented in the Appendix A. They include the different activities involved in the vaccination campaign such as vaccine purchase, vaccine transportation, vaccine storage, vaccine field delivery, sensitization, supervision, and coordination.
*Vaccine purchase cost* = *Single vaccine price* × *Number of doses used*(4)
*Vaccine transportation cost* = *Importing cost* + *Mean of transportation cost* + *Fuel cost* + *Personnel cost* + *Other transport cost*(5)
*Vaccine storage cost* = *Cold-room cost* + *Refrigerator cost* + *Power cost* + *Gas cost*(6)
*Vaccine field delivery cost* = *Personnel cost* + *Mean of transportation cost* + *Fuel cost* + *Material of vaccination cost* + *vaccine wastage* + *Other costs*(7)
*Sensitization cost* = *Personnel cost* + *Media board cast cost* + *Material and other cost*(8)
*Supervision cost* = *Personnel cost* + *Material cost* + *Other cost*(9)
*Training and meetings cost* = *Personnel cost* + *Material cost* + *Other cost*(10)
*Coordination cost* = *Personnel cost* + *Reporting cost* + *Other cost*(11)

#### 2.3.3. Cost Calculation by Type of Inputs

The details of calculation by inputs are presented in the Appendix A. They include the different inputs used for the vaccination campaign implementation such as personnel, material and logistics, overhead, and wastage.
*Personnel cost* = *personnel expense for transport* + *personnel expense for field delivery* + *sensitization* + *training and meeting* + *supervision* + *coordination*(12)
*Material and logistic cost* = *vaccine transport material* + *storage material* + *field delivery material* + *sensitization material* + *supervision material* + *training and meetings material* + *coordination material*(13)
*Overheads* = *Mean of transportation maintenance* + *Cold chain equipment maintenance* + *Fuel + Electricity (power, gas)* + *import and freight* + *Media broadcast* + *Other (ice, phone call, refreshment, farmer payment, toll fees)*.(14)
*Vaccine wastage cost* = *No of doses delivered* – *No of animal vaccinated* – *No of doses remaining unused in cold chain*(15)

Note that only the wastage at the field level (communal and private vets) has been included, the wastage elsewhere being negligible.

#### 2.3.4. Cost Calculation by the Nature of Cost

We considered fixed costs and variable costs. The details of calculation are presented in the Appendix A.
*Total variable cost* = *variable cost in vaccine purchase* + *variable cost in vaccine transportation* + *variable cost in vac-cine field delivery* + *variable cost in sensitization* + *variable cost in supervision* + *variable cost in training and meetings* + *variable cost in coordination*(16)
*Total fixed cost* = *fixed cost in vaccine purchase* + *fixed cost in vaccine transportation* + *fixed cost in vaccine field delivery* + *fixed cost in sensitization* + *fixed cost in supervision* + *fixed cost in training and meetings* + *fixed cost in coordination*(17)

#### 2.3.5. Total Cost Calculation by Vaccine Distribution Channel

The total of each cost component all levels combined was obtained using this method Appendix A:*Public total cost* = *central cost* × *0.6* + *regional cost* + *provincial cost* + *communal cost*(18)
*Private total cost* = *central cost* × *0.4* + *private veterinarian’s cost*(19)

In general, the number of SR vaccinated by the public sector and the private channel represents, respectively, 60% and 40% of the total SR vaccinated in the country. So, we assumed that the cost at the central level is shared following the same proportions through the 2 channels, respectively, 60% (0.6) and 40% (0.4).

The sum of all component costs was made to obtain the total cost of the PPR vaccination.

## 3. Results

### 3.1. PPR Vaccination Cost by Type of Activities

The average PPR vaccination unit cost is XOF 169 in the public distribution channel and XOF 103 in the private distribution channel. The repartition of this cost based on the type of activities (Table 1, Appendix A) shows that the most significant part of the cost comes from the vaccine field delivery, the sensitization, and the vaccine transport, all of which represent more than 78% (XOF 131) of the unit vaccination cost in the public channel and 75% (XOF 79) in the private channel.

### 3.2. PPR Vaccination Cost by Type of Inputs

Table 2 and Appendix A show that personnel is the input with the highest cost in the PPR vaccination representing 65% both in the public and the private channel. This is followed by the overheads that represent 17% (XOF 29) in the public channel and 18.8% (XOF 19) in the private channel. The vaccine wastage in the public channel (XOF 2) is more significant than in the private channel (XOF 0.21).

### 3.3. PPR Vaccination Cost by the Nature of the Cost

Variable costs represent 54% and 74% of the PPR vaccination cost in the public channel and the private channel, respectively (Table 3 and Appendix A). The percentage of fixed costs in the public channel is higher (46%) compared to the private channel (26%).

### 3.4. PPR Vaccination Cost Repartition by Level of Distribution

Combining all categories of costs (activities, inputs, and nature of costs), we observed that in the public channel, most of the PPR vaccination cost comes from the vaccinators (XOF 111; 66%) followed by the central veterinary services (XOF 45; 27%). The intermediate level (regional and provincial) is incurring a lower cost (Table 4, Appendix A). In the private channel, the field actors (private vets) contribute 62% of the cost (XOF 65) (Table 5 and Appendix A).

## 4. Discussion

### 4.1. PPR Vaccination Strategy Framework in Burkina Faso

Mass vaccination campaigns are implemented through a PPP approach [16]. The public veterinary services lead and orient the entire process. Once the vaccines are imported from abroad (mainly Jordan and Morocco) into the country, they are stored in the cold room at the central veterinary services (in Ouagadougou, Burkina Faso) until distributed. The vaccine distribution channels occur concomitantly during the campaign through a public–private partnership (PPP) (Figure 2). This partnership is important during the mass vaccination campaign because it enables vaccinators to reach out to many locations, thus increasing vaccination coverage.

Public channel: Here, the central government, through its regional agencies, is responsible for the distribution of the vaccines, as well as the vaccination of animals in areas where their staff is available. Most of times in such areas, there are no established private veterinarians. Vaccines are distributed successively from central veterinary services called General Directorate of Veterinary Services (Direction Générale des Services Vétérinaires -DGSV) to 13 regional services called Regional Directorate of Animal and Fisheries Resources (Direction Régionale des Resources Animales et Halieutiques -DRRAH), 45 provincial services called Provincial Directorate of Animal and Fisheries Resources (Direction Provinciale des Resources Animales et Halieutiques -DPRAH), and 351 communal services representing the vaccinators including the staff of the Veterinary Posts (Poste vétérinaire -PV), Livestock Technical Support Area (Zone d’Appui Technique en Elevage -ZATE), and Livestock Technical Support Unit (Unité d’Appui Technique en Elevage -UATE).Private channel: The distribution of the vaccine through the private channel is carried out by the veterinarians who hold the sanitary mandate. The sanitary mandate consists of an official assignment of the vaccination act to a private veterinarian by the government in a given area at the commune and province levels. The veterinarian who holds a sanitary mandate is called a “mandataire”. He has the sole right to vaccinate to get paid in the assigned area. In 2020, only 16 private veterinarians held sanitary mandates throughout the country. During the 2020 campaign, the number of animals vaccinated by the private sector represented 40% of the total number of small ruminants vaccinated in the country [20].

### 4.2. PPR Vaccination Cost

The average cost per vaccinated animal is XOF 169 (USD 0.3) and XOF 103 (USD 0.18) through the public and private channels, respectively. Similar studies on PPR vaccination have been carried out in Senegal, Nigeria, Ethiopia, and Somali (a region of Ethiopia). In Senegal, the estimated cost per SR vaccinated against PPR was between XOF 110 (USD 0.19) and XOF 187 (USD 0.33) according to different scenarios based on the productivity of vaccinators without distinction of channel type [22]. In Nigeria, this cost amounted to XOF 127 (USD 0.22) [23]. In Ethiopia, it was estimated at XOF 55 (USD 0.1) and XOF 110 (USD 0.19), respectively, in the pastoral and the agropastoral systems [24]. However, the estimations did not include the costs of field supervision and training of the vaccinators as in the case of Senegal and Burkina Faso. In the Somali region of Ethiopia, the cost of vaccinating a small ruminant was estimated to be between XOF 22 (USD 0.04) and XOF 43 (USD 0.08). Unlike the present study, the cost of the vaccine as well as the cost of the storage of the vaccine was not included in the calculation; this cost is, however, not negligible.

#### 4.2.1. In Terms of Activities

The field delivery of vaccines represented more than half of the vaccination cost (51% and 52%). The other important costs were sensitization (17% and 10%) and vaccine transportation (10% and 12%), respectively, through the public and private channels. In Ethiopia, vaccine field delivery represented 65% and 17% in the pastoral and the agropastoral systems, respectively. However, the cost of sensitization remains lower than in this study, i.e., 1.3% and 2.4%, respectively, in the pastoral and the agropastoral systems. This could be explained by the fact that no launching ceremony cost was considered as is the case for Burkina Faso in our study. In the case of Ethiopia, it is rather the farmer’s time for vaccination that represented an important proportion of the cost, especially in the agropastoral system where it reaches up to 65% of the cost of the total vaccination cost. In our study, we did not include the cost associated with the farmer’s time to vaccinate because it was negligible. To have their animals vaccinated, most farmers travel very short distances or do not travel at all, as the vaccinator moves from door to door.

In our study, the vaccine transport cost is also important, and it is higher through the public channel due to the long vaccine supply chain from the central level to the communes. Each day, the vaccinators should transport the vaccine in a cold box from the office to the vaccination sites located in remote areas. This situation causes an important cost of fuel and ice for vaccine storage. The most important part of the transport is the fuel cost for refrigerated vans and vaccinators in the field.

The study shows that the cost of vaccine storage is higher in the public channel with XOF 8 (USD 0.01) than in the private channel with XOF 3 (USD 0.005). This may be due to the long chain of vaccine distribution in the public channel: central, regional, provincial veterinary services, public vaccinators and farmers, while the chain is shorter for the private channel: central veterinary services, private veterinarians, and farmers. The vaccine spends time at each level of the chain resulting in the cost of electricity, depreciation, and maintenance of cold chain equipment. In the case of Ethiopia, the cost of conservation is quite low regardless of the farming system (less than 0.05%). This could be explained by the low cost of a kilowatt-hour (Kwh) in Ethiopia which is around XOF 3.6 (USD 0.006) while it ranges from XOF 88 (USD 0.15) to XOF 165 (USD 0.29) depending on the level of consumption in Burkina Faso. Added to this is the use of gas by some field staff (vaccinators) and whose loading cost is expensive with XOF 6,000 (USD 10.6) per bottle for 15 days of use.

#### 4.2.2. In Terms of Inputs

The inputs which have more weight in the total PPR vaccination cost are personnel costs (65% and 65%) and overheads (17% and 19%), respectively, in the public and the private channels. The cost components are higher in the public channel. Personnel costs include salaries and per diems for transport, vaccination, sensitization, supervision, training, meetings, and coordination. Additionally, the cost of vaccinators reduced to the number of animals vaccinated is more important in the public channel than in the private channel. Indeed, in the public channel, each vaccinator is paid a fixed amount of XOF 150,000 (USD 265) per campaign regardless of the number of animals vaccinated, knowing that the number of animals vaccinated per team of two vaccinators is estimated at 11,191 in the public channel per campaign. In the private sector, an amount of XOF 50 (USD 0.09) is paid to private veterinarians per vaccinated animal knowing that the total number of animals vaccinated by a team of two vaccinators is 14,045 per campaign. Private vaccinators seem to be more productive based on this factor. In the study carried out by [22] in Senegal, personnel costs ranged from 27% to 53% depending on the productivity of vaccinators (number of animals vaccinated per day), the share of personnel costs was reduced with increasing productivity. In Burkina Faso, the productivity of vaccinators could be influenced by the vaccination system. In fact, in the absence of vaccination parks and the conditions for rearing SR, door-to-door vaccination is carried out most of the time. This forces the agents to travel long distances to finally vaccinate a low number of animals with all the costs that this generates (fuel, depreciation, and maintenance of motorcycles).

Overheads include maintenance costs for transportation and cold chain equipment, fuel, electricity, radio communication, ice, and phone calls. Among the overheads, the costs related to supply in the field are particularly important. The daily fuel costs for each team are estimated at XOF 1,750 (USD 3) for the public channel and XOF 2000 (USD 3.5) for the private channel. This cost represents only 1% of the total cost of vaccination in Senegal, all scenarios combined [22]. However, this only includes the maintenance costs of cold chain equipment, electricity, and the maintenance of means of transportation.

The study shows that vaccine wastage is greater in the public channel with XOF 2 (USD 0.004) representing 1% when compared with the private channel with XOF 0.21 (USD 0.0004) representing 0.2%). As mentioned above, private veterinarians are paid proportionally to their level of performance, which leads them to rationalize vaccine wastage to avoid loss of money.

#### 4.2.3. In Terms of the Nature of the Costs

Variable costs take a larger part than fixed costs in both public and private distribution channels, around 54% and 74%, respectively. There is an important difference between the contributory shares of fixed costs in the public channel (46%) and in the private channel (26%). These costs do not vary depending on the number of animals vaccinated. They consist of the cost of vaccine storage, sensitization, supervision, capacity building, and coordination. To this must be added the personnel costs (per diem) of public sector vaccinators. They are paid at a fixed amount of XOF 150,000 (USD 265) for the implementation of the campaign while the private veterinarians are paid according to the number of animals vaccinated for an amount of XOF 50 (USD 0.09) per animal making the variable costs much higher in the private channel. This calls for better management of structural factors to optimize and rationalize the cost of vaccination in the public system.

#### 4.2.4. Distribution of the Vaccination Cost along the Vaccine Supply Chain

The results show that the levels which bear more of the cost of vaccination are the central level and the field level, the field level being more important. This is valid for the public channel and the private channel. This makes sense since the vaccine spends more time in these two levels causing costs. The field level bears more costs because most of the activities as part of the vaccination strategy take place in the field.

## 5. Conclusions

The results show that the cost of PPR vaccination through the public channel is higher compared to the private channel.

The vaccine delivery to the field, sensitization, and vaccine transportation represents a higher proportion of the vaccination cost. Personnel and overheads represent the highest proportion of the input cost of the vaccination. The variable costs are more important compared to the fixed cost in both public and private channels, but the cost is higher in the private channel.

The study shows that the efficiency of vaccinators is an important factor in the cost of vaccination. Productivity can be enhanced by improving the field vaccination cost efficiency that would lead to an increase in the number of animals vaccinated per day by the vaccination teams. We also recommend finding ways for combining the PPR vaccination campaign with other vaccination campaign(s) which is/are technically suitable to reduce the cost of vaccination.

These results inform veterinary services’ investment decision options for better allocation of resources. This, in turn, enhances the efficiency of control of PPR and other livestock diseases in Burkina Faso and in the Sahel, in general.

## Figures and Tables

**Figure 1 animals-12-02152-f001:**
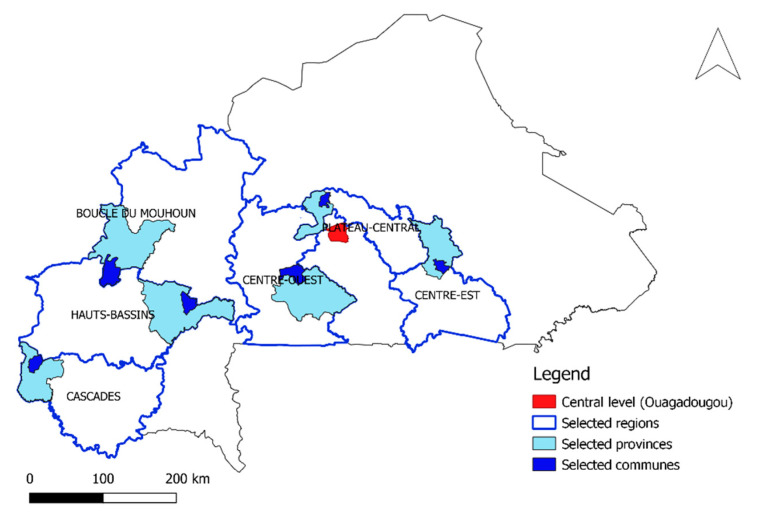
Map of Burkina Faso with the study areas (mapped by the first author using QGIS).

**Figure 2 animals-12-02152-f002:**
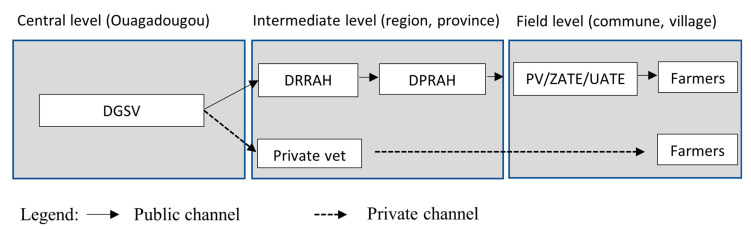
PPR vaccine distribution channels in Burkina Faso.

**Table 1 animals-12-02152-t001:** PPR vaccination cost by activities.

Type of Activities	Mean Cost (XOF) Per Dose	
	Public Channel	Private Channel
Vaccine purchase	14 (8%)	10 (9%)
Vaccine transport	17 (10%)	13 (12.5%)
Vaccine storage	8 (5%)	3 (3%)
Vaccine field delivery	86 (51%)	54 (52.5%)
Sensitization	28 (17%)	11 (10%)
Supervision	7 (4%)	3 (3%)
Training and meetings	5 (3%)	3 (3%)
Coordination	3 (2%)	7 (7%)
Total	169 (100%)	103 (100%)

**Table 2 animals-12-02152-t002:** PPR vaccination cost by inputs.

Type of Inputs	Mean Cost (XOF) Per Dose	
	Public Channel	Private Channel
Personnel	110 (65.4%)	66 (64.5%)
Material and logistic	14 (8.1%)	7 (7.2%)
Vaccine	14 (8.5%)	10 (9.3%)
Vaccine wastage	2 (1%)	0.21 (0.2%)
Overheads	29 (17%)	19 (18.8%)
Total	169 (100%)	103 (100%)

**Table 3 animals-12-02152-t003:** PPR vaccination cost by inputs.

Nature of the Costs	Mean Cost (XOF) Per Dose	
	Public Channel	Private Channel
Fixed cost	78 (46%)	27 (26%)
Variable cost	90 (54%)	74 (74%)
Total	169 (100%)	103 (100%)

**Table 4 animals-12-02152-t004:** Percentage contribution of level distribution to PPR vaccination cost in public distribution channels in Burkina Faso in 2020.

Cost Category	Central	Regional	Provincial	Public Vaccinators	Total
Mean cost per dose (XOF)	45.2	2.8	9.5	111.1	169
Percentage	26.8	1.7	5.6	65.9	100

**Table 5 animals-12-02152-t005:** Percentage contribution of level distribution to PPR vaccination cost in private distribution channels in Burkina Faso in 2020.

Cost Category	Central	Private Veterinarians	Total
Mean cost per dose (XOF)	39	64	103
Percentage	38	62	100

## Data Availability

Not applicable.

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
