# Peer review of "Peste des Petits Ruminants (PPR) Vaccination Cost Estimates in Burkina Faso"

_animals, 2022, doi:10.3390/ani12162152_

Round 1
Reviewer 1 Report
I would have liked the paper to mention that its results sit within those of previous studies of PPR vaccination costs. This provides support for their veracity and relevance. In fact a table comparing these results and other published results would be useful. And how do the PPR results compare with the costs of rinderpest vaccination years ago? Favourably, I think.
There is one section of the M & M that I feel needs clarification.
2.3.5. Total cost calculation by vaccine distribution channel
Public cost - central costs *0.6 + etc +etc
private cost + central cost *0.4 +etc.
I do not have access to Appendix D. The difference between 0.4 and 0.6 as a modifying factor is very significant I feel. Please explain why you have chosen these values?
Reviewer 2 Report
The authors present results from a study assessing and describing the costs associated with PPR vaccine delivery to sheep and goats in Burkina Faso based on the 2020 PPR vaccine campaign data. Appropriate comparisons are made to other similar studies and the content is important and relevant to efforts to improve the effectiveness and efficiency of the FAO/WOAH PPR Global Eradication Plan.
The difference in costs associated with PPR vaccine delivery through the public and private sector is calculated and methods adequately described. The results and discussion could be strengthened with inclusion of information from the key informant interviews on the factors that led to the selection of public veterinarians vs private veterinarians for vaccine distribution if those data are available.
A few minor suggestions for consideration:
Line 63: Change "SR's trade" to "SR trade"
Line 118: Typo "therteen" to "thirteen"
Line 147: "That includes" to "They included"
Line 286: Missing "." at end of sentence and formatting gap
Line 322: Change "don't" to "do not"
Line 408: Change "way" to "ways"
